# YKL-40 in Virus-Associated Liver Disease: A Translational Biomarker Linking Fibrosis, Hepatocarcinogenesis, and Liver Transplantation

**DOI:** 10.3390/ijms26199584

**Published:** 2025-10-01

**Authors:** Jadranka Pavicic Saric, Dinka Lulic, Dunja Rogic, Stipislav Jadrijevic, Danko Mikulic, Tajana Filipec Kanizaj, Nikola Prpic, Laura Karla Bozic, Ivona Adamovic, Iva Bacak Kocman, Zrinka Sarec, Gorjana Erceg, Mirta Adanic, Petra Ozegovic Zuljan, Filip Jadrijevic, Ileana Lulic

**Affiliations:** 1Solid Organ Transplant Unit, Department of Anesthesiology, Intensive Care and Pain Medicine, Clinical Hospital Merkur, Zajceva 19, 10000 Zagreb, Croatia; nikolaprpic2@gmail.com (N.P.); blaura210@gmail.com (L.K.B.); adamovicivona96@gmail.com (I.A.); bacakkocmaniva@gmail.com (I.B.K.); krznaricz@yahoo.com (Z.S.); gorjanaerceg@gmail.com (G.E.); mirtaadanic@gmail.com (M.A.); petraoze@gmail.com (P.O.Z.); ileanalulic@gmail.com (I.L.); 2Immediate Medical Care Unit, Saint James Hospital, SLM-1030 Sliema, Malta; dinka.lulic@gmail.com; 3Department of Medical Biochemistry and Hematology, Faculty of Pharmacy and Biochemistry, A. Kovacica 1, 10000 Zagreb, Croatia; predstojnik.lab@kbc-zagreb.hr; 4Department of Laboratory Diagnostics, University Hospital Centre Zagreb, Kispaticeva 12, 10000 Zagreb, Croatia; 5Solid Organ Transplant Unit, Department of Surgery, Clinical Hospital Merkur, Zajceva 19, 10000 Zagreb, Croatia; stipislavjadrijevic@gmail.com (S.J.); mikulicdanko@gmail.com (D.M.); 6Solid Organ Transplant Unit, Department of Gastroenterology, Clinical Hospital Merkur, Zajceva 19, 10000 Zagreb, Croatia; tajana.filipec@gmail.com; 7School of Medicine, University of Mostar, Zrinskog Frankopana 34, 88000 Mostar, Bosnia and Herzegovina; fjadrijevic1@gmail.com

**Keywords:** hepatitis, liver cirrhosis, tumor biomarkers, hepatocellular carcinoma, liver transplantation

## Abstract

Virus-associated hepatocellular carcinoma (HCC) remains a major global health burden despite effective antiviral therapies. Chronic infection with hepatitis B (HBV), hepatitis C (HCV), and hepatitis D (HDV) promotes malignant transformation through overlapping pathways of fibrosis, immune dysregulation, and microenvironmental remodeling. YKL-40, a glycoprotein secreted by hepatic stellate cells, hepatocytes under stress, macrophages, and endothelial cells, has emerged as a marker that reflects stromal activation rather than direct hepatocyte injury. Its expression is reinforced by profibrotic and angiogenic circuits, and circulating concentrations correlate with advanced fibrosis, residual risk after viral suppression, and oncologic outcomes. This review synthesizes current evidence on YKL-40 across HBV, HCV, and HDV cohorts, with emphasis on its role in bridging molecular mechanisms to clinical applications. We examine its utility in non-invasive fibrosis assessment, longitudinal monitoring after antiviral therapy, and prognostic modeling in HCC. Particular attention is given to its potential in the liver transplant pathway, where YKL-40 may refine eligibility beyond morphology, inform bridging therapy response, and predict post-transplant recurrence or graft fibrosis. Remaining challenges include its lack of disease specificity, assay variability, and limited multicenter validation. Future integration of YKL-40 into multimarker, algorithm-based frameworks could enable risk-adaptive strategies that align surveillance and transplant decisions with the evolving biology of virus-associated liver disease.

## 1. Introduction

Hepatocellular carcinoma (HCC) is the most common form of primary liver cancer and accounts for a substantial proportion of global cancer mortality [1]. Many cases develop in the context of chronic viral hepatitis, particularly from persistent infection with hepatitis B virus (HBV), hepatitis C virus (HCV), or hepatitis D virus (HDV) [2,3]. Antiviral strategies such as nucleos(t)ide analogs for HBV and direct-acting antivirals (DAAs) for HCV have significantly improved therapeutic options. However, important challenges remain, with persistent disparities in diagnosis and screening, limited access to antiviral treatment in many regions, and the continuing risk of virus-associated HCC even after viral suppression or clearance [4,5].

The progression from chronic viral hepatitis to HCC arises through cumulative hepatocellular injury, with activation of hepatic stellate cells (HSCs) driving extracellular matrix (ECM) deposition and fibrosis, while chronic inflammation and dysregulated regenerative signaling promote oncogenic transformation [6,7].

Current serum biomarkers for HCC surveillance, including alpha-fetoprotein (AFP) and des-gamma-carboxy prothrombin (DCP), have limited diagnostic performance, particularly in early-stage disease or in tumors lacking typical secretory phenotypes [8]. In virus-associated disease, where virologic control reduces inflammatory activity but does not abolish carcinogenic risk, biologically informative biomarkers are needed to refine risk-targeted surveillance, enable earlier detection and response assessment (including molecular residual disease), and complement morphologic criteria along the liver transplantation (LT) pathway (candidate selection, bridging/downstaging, post-LT surveillance). Consequently, current work spans complementary biomarker classes, such as tumor-derived proteins (e.g., glypican-3 and osteopontin), liquid biopsy analytes (circulating tumor DNA and microRNAs, including extracellular vesicle cargo), and extracellular vesicles themselves, which probe oncoprotein expression, genomic/epigenetic alterations, and vesicle-mediated stromal-immune crosstalk [9,10,11,12,13]. An overview of conventional and emerging biomarkers across the viral hepatitis-HCC-LT continuum is summarized in Table 1.

YKL-40 (chitinase-3-like protein 1, CHI3L1) is a glycoprotein produced by activated HSCs, macrophages, endothelial cells, and stressed hepatocytes. By engaging ECM turnover and inflammatory-angiogenic signaling, it serves as a readout of stromal activation central to virus-associated carcinogenesis [14]. Circulating concentrations correlate with fibrosis stage, aggressive tumor biology, and adverse outcomes in chronic liver disease and HCC, thereby complementing tumor-derived markers and imaging [15,16]. Accordingly, a structured evaluation of YKL-40 across viral etiologies and along the LT pathway is warranted.

Despite these associations, the virus-specific contributions of YKL-40 and its translational potential in guiding LT remain incompletely understood. In this narrative review, we critically synthesize current evidence on YKL-40 in HBV-, HCV-, and HDV-associated disease, emphasizing its mechanistic basis, clinical applications, and implications for biologically adaptive decision-making across the fibrosis-carcinogenesis-transplant continuum.

## 2. Mechanistic and Clinical Relevance of YKL-40 in Virus-Associated Fibrosis and Hepatocarcinogenesis

### 2.1. HBV: Viral Persistence, Fibrogenic Signaling, and Oncogenic Remodeling

The persistence of HBV is maintained through covalently closed circular DNA (cccDNA), a stable episomal template that remains transcriptionally active despite antiviral suppression [17]. Continuous viral gene expression creates a state of chronic immune disturbance, with exhausted T cells, increased secretion of interleukin-6 (IL-6) and tumor necrosis factor-alpha (TNF-α), and expansion of regulatory immune subsets that blunt antiviral cytotoxicity [18,19,20]. These signals converge on HSCs, where profibrogenic mediators such as transforming growth factor-beta (TGF-β), connective tissue growth factor (CTGF), and hepatocyte-derived exosomes initiate ECM deposition and architectural remodeling [21,22].

Sustained stimulation gradually reprograms stellate cells into a fibrogenic state characterized by enhanced collagen synthesis and reduced matrix degradation, supported by contractile and proliferative changes that consolidate their myofibroblastic phenotype [23,24,25,26,27]. Scar tissue becomes further stabilized through collagen cross linking by lysyl oxidase-like 2 (*LOXL2*), which limits the potential for reversal [28]. Insights from single-cell transcriptomic studies add nuance, showing that activation is not uniform across the liver but varies by zone, with periportal and pericentral stellate subsets contributing in distinct ways to the fibrotic landscape [29,30]. This heterogeneity has clinical significance because fibrosis may progress silently despite normal alanine aminotransferase (ALT) levels [31]. In this context, circulating YKL-40 has emerged as an informative biomarker, with elevated levels reported in HBV cohorts with advanced fibrosis and potential utility for staging and disease monitoring [32,33].

As fibrosis advances, the altered microenvironment facilitates oncogenic transformation. Epigenetic changes such as aberrant DNA methylation and dysregulated noncoding RNAs weaken tumor-suppressive programs, while HBV DNA integration activates oncogenes including telomerase reverse transcriptase (*TERT*) and cyclin E1 (*CCNE1*), promoting clonal expansion of genetically unstable hepatocytes [34,35,36,37]. These processes destabilize genomic integrity and create conditions permissive for malignant evolution [38].

HBV also promotes cancer through direct molecular drivers. The hepatitis B virus X protein (HBx) is central to this process. It impairs DNA repair, disrupts cell-cycle checkpoints, and alters chromosomal organization [39,40]. HBx can engage chromatin remodeling machinery such as SWI/SNF and displace Polycomb repressors, opening proto-oncogenic loci to transcriptional activation [41,42]. In parallel, interferon signaling and antigen presentation are dampened, and apoptotic thresholds are shifted via mitochondrial and death-receptor pathways, weakening immune surveillance [43,44]. Although no direct studies have linked HBx activity to YKL-40 expression, the fibrogenic and inflammatory environment fostered by HBV provides a biologically coherent basis for its up-regulation, emphasizing the need for studies that clarify its virus-specific contributions and translational potential. The key processes of viral persistence, fibrogenic signaling, and oncogenic remodeling in HBV are summarized in Figure 1.

### 2.2. HCV: Immune Polarization, Fibrotic Programming, and Inflammation-Driven Transformation

Unlike HBV, which achieves persistence through cccDNA reservoirs and integration into the host genome, HCV maintains chronic infection by evading innate immune recognition while sustaining replication in the hepatocyte cytoplasm [45,46]. Viral proteins such as NS3/4A and NS5A disable mitochondrial antiviral signaling protein (MAVS) and interfere with toll-like receptor (TLR)-dependent interferon cascades, weakening antiviral defenses mediated by retinoic acid-inducible gene I (*RIG-I*), interferon regulatory factor 3 (*IRF3*), and signal transducer and activator of transcription 1 (*STAT1*) [47,48].

Beyond immune evasion, HCV viral proteins actively drive profibrogenic signaling. Experimental studies have shown that NS5A and core proteins induce oxidative stress through NADPH oxidases and mitochondrial pathways, while simultaneously activating NF-κB and STAT3 signaling [49]. These pathways serve as established transcriptional activators of YKL-40, providing a mechanistic link between HCV protein activity and YKL-40 upregulation [49].

The YKL-40 promoter contains functional response elements for both *NF-κB* and *STAT3*, rendering it highly responsive to these signaling cascades [50]. HCV NS3 and NS5A proteins sustain STAT3 activation through prolonged phosphorylation, directly enhancing YKL-40 transcription and linking viral replication to persistent stromal activation and fibrogenesis [50].

Persistent viral replication and protein activity create an environment of chronic, low-grade inflammation rather than direct cytopathic damage. Virus-specific CD8^+^ T cells become progressively exhausted, with elevated expression of inhibitory receptors such as programmed cell death protein 1 (PD-1) and lymphocyte activation gene 3 (*LAG-3*) [51]. Regulatory T cells (Tregs) expand and further suppress cytotoxic responses, while Kupffer cells and infiltrating macrophages polarize toward a profibrogenic phenotype marked by IL-10, TGF-β, and arginase-1 [52].

These immune changes converge on HSCs, initiating ECM deposition and stromal remodeling. Reactive oxygen species (ROS), pro-inflammatory cytokines such as IL-1β, and chemokines like CXCL10 reinforce this fibrogenic circuit, creating a self-perpetuating inflammatory loop [53,54]. In this milieu, serum YKL-40 levels are consistently higher in HCV patients with advanced fibrosis, reflecting both matrix turnover and the intensity of macrophage-stellate cell crosstalk [55].

Macrophage polarization further stabilizes this fibrogenic axis. Under sustained IL-10 and TGF-β exposure, alternatively activated macrophages drive HSCs toward a myofibroblastic state characterized by increased collagen production, impaired ECM degradation, and expression of markers such as *ACTA2*, *COL1A1*, and *TIMP1* [56,57,58]. Kupffer cells and infiltrating macrophages amplify this circuit through secretion of *CCL2* and osteopontin, which recruit additional inflammatory and profibrogenic cells into perisinusoidal regions [59,60].

As fibrosis matures, structural cues reinforce its persistence. Increasing ECM stiffness enhances integrin-mediated activation of latent TGF-β, locking HSCs into a self-sustaining fibrogenic program [61]. This process is amplified by lysyl oxidase-like 2 (*LOXL2*), which promotes collagen crosslinking [62], while downregulation of matrix metalloproteinases (*MMPs*) limits turnover and enforces rigidity [63]. Insights from single-cell transcriptomics reveal zone-specific heterogeneity of HSC activation, with central vein-associated subsets showing dominant collagen production and distinct expression of *ACTA2*, remodeling enzymes, and chemokine receptors [64]. Endothelial signaling and oxygen gradients further refine these spatial responses, integrating immune, vascular, and metabolic cues into the evolving fibrotic architecture [65]. Figure 2 summarizes the sequence of immune dysfunction, fibrotic remodeling, and YKL-40 induction in HCV-associated liver disease.

### 2.3. HDV: Intensified Immunopathology and Fibrotic Escalation

HDV depends on the hepatitis B surface antigen (HBsAg) for assembly and propagation, making HBV co- or superinfection a prerequisite for its persistence [66]. Clinically, HBV/HDV coinfection follows a more aggressive course than HBV alone, with accelerated progression to cirrhosis, earlier decompensation, and higher risk of HCC [67].

The large hepatitis D antigen (L-HDAg) is a central driver of this pathogenic phenotype [68]. Beyond its direct viral functions, L-HDAg promotes an inflammatory and profibrogenic milieu within the liver. Kupffer cells and infiltrating monocytes adopt activated states and release cytokines such as TNF-α and IL-1β, which sustain stellate-cell activity and drive collagen I accumulation [68]. Together these mechanisms provide a foundation for the rapid fibrotic trajectory observed in HBV/HDV coinfection.

Additional studies show that L-HDAg also promotes oxidative stress through nicotinamide adenine dinucleotide phosphate (NADPH) oxidase and activates STAT3 and nuclear factor kappa-light-chain-enhancer of activated B cells (NF-κB) signaling [69]. It further disrupts endoplasmic reticulum homeostasis, triggers unfolded-protein-response pathways, and perturbs proteostasis, thereby maintaining a chronically inflamed hepatic niche reinforced by type I interferon activity [69]. These signals accelerate stellate-cell activation and matrix deposition, deepening the fibrotic response.

Clinical evidence supports these mechanistic observations, demonstrating that fibrosis progresses more rapidly in HBV/HDV coinfection compared with HBV monoinfection, in line with the heightened inflammatory burden [70]. Adaptive immunity is also reshaped. Virus-specific CD8^+^ T cells gradually acquire features of exhaustion, regulatory pathways expand, and cytotoxic lymphocytes lose functional capacity. This imbalance limits viral control while amplifying hepatocellular injury, hastening cirrhosis and elevating oncogenic risk compared with HBV alone [71,72].

Direct evaluation of YKL-40 in HDV infection is not yet available. Even so, the profibrotic and inflammatory profile of HBV/HDV coinfection creates a biologically plausible setting for its up-regulation. Evidence from HBV and HCV cohorts, where higher YKL-40 levels correlate with advanced fibrosis and adverse outcomes, provides a strong rationale for testing whether similar associations are present in HDV. Addressing this gap will be important for determining whether YKL-40 can serve as a clinically useful biomarker in this uniquely aggressive viral context. Future research could employ multi-omics approaches, such as single-cell transcriptomics of HDV-infected hepatocytes or stellate cells, to identify cellular sources and regulatory networks driving YKL-40 expression. In parallel, clinical cohort studies with paired serum and liver biopsy data could define the diagnostic and prognostic performance of YKL-40 in HBV/HDV coinfection, establishing its role as a potential biomarker across disease stages. Figure 3 provides a visual summary of the intensified immunopathology and rapid fibrotic escalation characteristic of HBV/HDV coinfection, illustrating how L-HDAg-driven immune dysregulation, oxidative stress, and stellate-cell activation converge on YKL-40 induction and heightened oncogenic risk.

### 2.4. Cross-Etiology Synthesis: YKL-40 as a Read-Out of Stromal Activation

HBV, HCV, and HDV maintain persistence through different virological strategies, yet over time they drive the liver toward a similar endpoint in which the hepatic microenvironment becomes fibrotic and immunologically altered, creating conditions that favor malignant progression [73,74,75]. Within this shared landscape, YKL-40 reflects stromal activation that connects fibrosis with oncogenic risk, providing a common signal across viral contexts [76].

The convergent and virus-specific pathways underlying YKL-40 induction are depicted in Figure 4 and summarized in Table 2, highlighting how HBV, HCV, and HDV engage distinct viral drivers that ultimately activate overlapping profibrotic and inflammatory cascades.

## 3. YKL-40 as a Translational Biomarker in Virus-Associated Liver Disease

### 3.1. Molecular Basis of YKL-40 Expression

YKL-40 stands out among candidate biomarkers because it is produced by different cell populations that act together in the chronically injured liver. This explains why it is consistently measurable in both tissue and circulation and why its signal is not limited to hepatocyte injury alone [77,78].

The strongest evidence for YKL-40 production comes from HSCs, which secrete it as they adopt a matrix-producing phenotype during fibrogenesis [79,80]. Hepatocytes also contribute when exposed to stress, particularly at the edges of fibrous septa, where staining is most pronounced in HDV, followed by HCV and HBV [79]. Macrophages add further expression during inflammatory activation [77,78], while endothelial and epithelial cells increase secretion under hypoxia or pro-angiogenic conditions [77,80]. These diverse sources position YKL-40 as a multicellular stress response that reflects how parenchymal, stromal, and vascular compartments adapt under chronic injury.

Its induction is reinforced by inflammatory and stress-responsive circuits. IL-1β and IL-6 activate STAT3 and NF-κB, embedding YKL-40 in transcriptional programs that remain active long after acute injury subsides [50]. Oxidative stress, matrix stiffening, and hypoxia further sustain expression, which explains why serum concentrations remain elevated in advanced fibrosis even when ALT values normalize [81].

Secreted YKL-40 then links these upstream signals to structural remodeling. Binding to interleukin-13 receptor subunit alpha-2 (IL-13Rα2) together with transmembrane protein 219 activates extracellular signal-regulated kinase (ERK), protein kinase B (AKT), and Wingless-related integration site/β-catenin (Wnt/β-catenin) signaling [82]. In addition to being a marker of fibrosis, YKL-40 actively participates in a positive feedback loop that amplifies fibrogenesis. Upon secretion, YKL-40 binds to IL-13Rα2 in complex with TMEM219, triggering downstream MAPK and PI3K/AKT signaling cascades. These pathways promote HSC proliferation and collagen synthesis, reinforcing ECM deposition and stiffening. Increasing matrix stiffness in turn sustains stellate cell activation and further upregulates YKL-40 expression, creating a self-perpetuating cycle of fibrosis progression [82,83]. Figure 5 illustrates these interconnected mechanisms, showing how viral proteins, cytokines, and oxidative stress activate multiple liver cell populations to secrete YKL-40, which in turn drives fibrosis, angiogenesis, and ECM stiffening through MAPK and PI3K/AKT signaling pathways.

This mechanistic loop explains why circulating YKL-40 levels closely track disease severity and remain elevated even when upstream injury markers, such as ALT, return to normal. Interactions with syndecan-1 and integrins αvβ3 and αvβ5 amplify motility and ECM turnover in endothelial and stromal cells [83]. These overlapping receptor pathways align with the consistent observation that YKL-40 is concentrated at fibrotic and angiogenic interfaces in diseased tissue [79,84].

Functional studies illustrate the consequences of this biology. In stellate cells, YKL-40 stimulates proliferation and collagen synthesis, accelerating matrix accumulation [78,79]. In endothelial models, it enhances tube formation and vascular endothelial growth factor (VEGF) expression, supporting the angiogenic remodeling characteristic of cirrhosis [83]. Neutralization experiments demonstrate that blocking YKL-40 can reduce tumor vascularization, confirming its role at the tumor-stroma interface [83]. In immune compartments, YKL-40 promotes macrophage recruitment and polarization toward an alternatively activated (M2-like) phenotype, reshaping cytokine balance and matrix metalloproteinase activity in ways that favor tolerance and invasive growth [76,85].

Virus-specific data strengthen these mechanistic insights. In HCV infection, YKL-40 supports hepatocyte viability and profibrogenic cytokine release, and there is evidence that it may even facilitate viral replication [81]. In HBV and HCV cohorts, circulating levels rise in step with fibrosis stage and remain elevated when hepatocyte injury markers decline [77,81]. Broader reviews of HSC activation and chitinase-like proteins in cancer highlight YKL-40 as a central driver of stromal remodeling that connects fibrotic progression with oncogenic potential [79,84].

Taken together, these findings show that YKL-40 is not just a bystander of tissue damage but a multicellular stress signal sustained by cytokines and environmental inputs. Its effects on fibrogenesis, angiogenesis, and immune adaptation explain why it is stable in circulation and provide the rationale for testing its translational value across HBV, HCV, and HDV populations and along the transplant pathway.

### 3.2. Clinical and Translational Applications of YKL-40

Evidence now quantifies how much information YKL-40 can add to day-to-day decisions. A recent meta-analysis that pooled seventeen liver fibrosis studies reported areas under the receiver operating characteristic curve (AUC) of 0.91 for advanced fibrosis and 0.87 for severe fibrosis with corresponding sensitivities near 0.80 and specificities near 0.85 across cohorts, which places YKL-40 in a performance range suitable for clinical triage rather than exploratory use [86]. The question then becomes where such a signal changes management. Histology-driven work in chronic hepatitis B shows that a considerable share of patients who meet biochemical definitions of normal ALT still carry significant necroinflammation or stage F2 or higher. In these cohorts the proportion with significant histological disease remains near a quarter to a third even when stricter upper limits of normal are applied, which creates a clear use case for a marker that can prioritize elastography or biopsy when standard biochemistry is quiet [87]. A second study focused on HBeAg-positive hepatitis B with normal ALT reached the same practical conclusion by detailing how often clinically silent activity appears on biopsy, which again supports targeted non-invasive testing rather than routine deferral of staging [88].

A move from static to longitudinal use further clarifies how to integrate YKL-40. In chronic hepatitis B, models that incorporate repeated measurements classify cirrhosis with an AUC of 0.939 and capture change over follow up, which is the type of performance that supports interval testing in clinics that already run serial algorithms for inactive carriers and for treated patients [88]. In HBeAg-negative hepatitis B, single-timepoint YKL-40 discriminates minimal from significant fibrosis with an AUC of 0.818 and with sensitivity and specificity around eighty and seventy percent. It also aligns with fibrosis indices such as FIB-4 and the gamma-glutamyl transpeptidase to platelet ratio, which allows simple cross-checks during routine visits without adding imaging at every timepoint [89]. At the level of pooled evidence, a meta-analysis across viral hepatitis datasets reported overall sensitivity and specificity near seventy four and seventy six percent, which provides a benchmark for centers that are considering adoption and want to compare their own operating points against external data [90].

Therapy studies show when the marker should be rechecked. In HCV infection, YKL-40 correlates with liver stiffness at baseline and detects cirrhosis with an AUC of 0.939. After direct-acting antivirals it falls by about one fifth within months, yet levels in many patients remain above those of healthy comparators. Untreated cirrhotics trend upward over the same horizon, which maps the marker to real trajectories rather than to a single post-treatment snapshot and supports planned retesting after cure in patients who remain at risk [90]. Host factors also shape downstream outcomes. In patients who cleared hepatitis C with direct-acting antivirals, common variants in the YKL-40 locus and an intergenic site identified groups with higher HCC risk after virological cure. The risk was greatest when more than one locus carried the at-risk genotype, which points to a host component that can be considered alongside serum measurement when long-term surveillance is planned [91].

Oncologic endpoints define where YKL-40 influences decisions beyond fibrosis staging. A meta-analysis of forty one cohorts across solid tumors showed that elevated serum YKL-40 associated with poorer overall survival with a pooled hazard ratio of 1.44, which sets an external expectation for effect size when the marker is used to refine prognosis [92]. HCC specific data mirror that pattern. After curative hepatectomy, higher preoperative YKL-40 independently predicted overall survival with adjusted hazard ratios in the range of 1.4 to 1.5, and a prognostic nomogram that added YKL-40 to tumor number, tumor size, the neutrophil to lymphocyte ratio, the international normalized ratio, and alpha-fetoprotein achieved areas under the curve near 0.75 in training and validation. A practical cut-point close to 199 ng per milliliter separated survival curves, which offers a starting threshold for programs that are designing preoperative risk conferences and postoperative follow up schedules [93].

Implementation details determine how reliable the readouts will be once YKL-40 is brought into routine use. Population data indicate that circulating levels rise with age, and that up to about one quarter of interindividual variation can be explained by common genetic polymorphisms in the YKL-40 gene. Laboratory practice also matters because most measurements use immunoassay formats with different reportable ranges and different sample handling requirements. These facts argue for age aware interpretation, for attention to serial change rather than single values, and for harmonized pre-analytical routines so that trajectories reflect biology rather than processing noise [94]. Broader oncology experience reaches a similar conclusion about standardization and about context. YKL-40 is elevated across many inflammatory and neoplastic states. It is therefore most useful when applied in defined hepatology pathways that already incorporate imaging and clinical scores and when its thresholds and sampling intervals are prespecified by etiology and treatment state [95].

These clinical studies show that YKL-40 has progressed from a descriptive marker of stromal activity to a tool with operational relevance. It uncovers fibrosis that escapes detection by routine biochemistry, it remains informative after antiviral cure when residual scarring drives long-term risk, and it strengthens prognostic models in HCC, particularly in patients without conventional tumor marker elevation. At the same time, its performance is influenced by age, genetic variation, and assay practice, underlining the importance of standardized protocols for future adoption. The convergence of these findings highlights why YKL-40 is now being considered not only for staging and surveillance but also for peri-transplant assessment, where the balance between fibrotic burden, oncologic risk, and immune adaptation becomes critical. To provide a comprehensive overview of its diagnostic performance, Table 3 summarizes key studies reporting receiver operating characteristic (ROC) values, optimal cut-off thresholds, and corresponding sensitivity and specificity of YKL-40 in HBV and HCV cohorts.

## 4. Pre- and Post-Transplant Applications of YKL-40 in HCC: From Eligibility to Risk Stratification

### 4.1. Informing Transplant Eligibility Beyond Morphology

Selection for LT in HCC has traditionally been determined by morphologic frameworks such as the Milan and University of California, San Francisco (UCSF) criteria, which define limits of tumor size and number [96]. These thresholds improved outcomes by restricting transplantation to patients with favorable imaging features, yet they cannot capture tumor biology or the state of the hepatic microenvironment, both of which drive recurrence risk after transplantation [97]. More recent allocation tools, including Metroticket 2.0 and AFP-based models, have added serological signals to extend predictive accuracy, but they remain limited in patients with low or discordant AFP expression [98,99].

YKL-40 provides a way to interrogate the biological state of the host rather than the morphology of the tumor [100]. In practice, this can sharpen decision-making at two critical junctures. First, in candidates who appear to meet morphologic thresholds, high YKL-40 may uncover a stromal environment already primed for recurrence, suggesting that eligibility should be reconsidered or that intensified bridging therapy is warranted before listing. Second, in patients undergoing downstaging, declining YKL-40 values can provide reassurance that the fibrotic and angiogenic drive has slowed, strengthening the case to proceed to transplantation. Data from TACE cohorts reinforce these scenarios, where serum YKL-40 predicted overall survival independently of AFP and retained value in AFP-negative phenotypes, which are precisely the cases where current allocation models are least informative [101]. Figure 6 provides a visual summary of how YKL-40 contributes to decision-making at different stages of the LT pathway, from pre-transplant eligibility to post-transplant surveillance.

### 4.2. Refining Pre-Transplant Risk Stratification

Once eligibility is confirmed, the next challenge is to refine biological risk before committing scarce donor organs [102,103]. Conventional stratification relies on imaging and response to bridging therapy, yet these assessments can miss the stromal activity that sustains recurrence beneath apparently stable radiology [104,105].

YKL-40 provides a complementary perspective by signaling microenvironmental activity that is otherwise invisible. Elevated levels correlate with larger tumor burden, vascular invasion, and reduced survival across HCC cohorts [106,107]. Trajectories of YKL-40 during bridging therapy add actionable value. Persistently high concentrations indicate that stromal activity continues and that recurrence risk is greater. Declining levels indicate that biological activity has slowed and provide stronger reassurance to proceed with transplantation.

The mechanistic basis for these associations has been described in experimental models, but their clinical implications can be summarized more directly. Table 4 illustrates how YKL-40 related pathways map onto outcomes that matter in the transplant setting, converting biological activity into markers of recurrence risk and treatment response.

### 4.3. Post-Transplant Recurrence Prediction and Integration into Biomarker Panels

Even with stringent selection criteria, 10–20% of LT recipients experience HCC recurrence, which emphasizes the limits of morphology-based models [108,109]. Recent studies now provide LT-specific evidence. In a large transplant cohort, elevated peri-transplant YKL-40 was linked with shorter overall and recurrence-free survival. The association was strongest in patients with myosteatosis, a systemic frailty phenotype that is invisible to conventional tumor markers [110]. These results identify YKL-40 as a clinically meaningful signal for peri-transplant risk stratification, particularly in patients who are AFP- or DCP-negative and therefore lack reliable tumor-derived markers [93].

The value of YKL-40 is not confined to recurrence alone. It also shows potential as a biomarker of graft fibrosis. In recipients with recurrent hepatitis C, serial measurements distinguished rapid from slow fibrosis progressors and tracked with histologic progression [111]. Genetic findings reinforce this plausibility. A functional promoter polymorphism in the YKL-40 gene was associated with higher circulating levels, more severe rejection, and faster fibrosis progression after transplantation [112]. These observations support a dual role for YKL-40 in the post-transplant setting, both as a predictor of oncologic recurrence and as a non-invasive marker of graft injury.

Implementation requires careful attention to context. YKL-40 is not disease specific, and elevations can reflect infection, rejection, or systemic inflammation, all of which are common in the early post-transplant course [94]. Laboratory factors also matter, since assay platforms vary in sensitivity and reproducibility, and recent high-dynamic-range immunoassays have highlighted challenges for cross-center comparison [113]. For the marker to be integrated into practice, multicenter validation is needed to confirm whether its addition to prognostic models such as RETREAT or MoRAL provides meaningful reclassification and supports risk-adapted surveillance strategies [114].

### 4.4. Future Directions

Future investigations will need to focus on translating YKL-40 from a descriptive biomarker into a clinically actionable tool across the spectrum of virus-associated liver disease. At earlier stages, longitudinal monitoring of YKL-40 offers a strategy to capture ongoing stromal remodeling and to assess fibrosis regression during antiviral or antifibrotic therapy. Large, prospective cohorts with standardized protocols are required to define biologically meaningful fluctuations, link them to histological and clinical outcomes, and establish its role in risk stratification and therapeutic planning.

As liver architecture becomes more disrupted, YKL-40 has the potential to signal transition toward decompensation before overt clinical manifestations arise. When incorporated into multivariable frameworks alongside established clinical variables, this biomarker could refine the prediction of complications such as variceal bleeding, ascites, and hepatic encephalopathy. Aligning surveillance and preventive interventions with molecular disease activity rather than static scores would enable timelier and more efficient use of resources.

Persistent stromal activation also contributes to tumor initiation, progression, and recurrence, even when viral replication is well controlled. Future efforts should explore integrating YKL-40 into composite biomarker signatures that merge stromal, immune, and tumor-derived signals with advanced imaging and circulating markers such as ctDNA and microRNAs. Such models could enhance early detection, recurrence prediction, and post-treatment monitoring, particularly for AFP-negative phenotypes and patients under surveillance following curative-intent therapies.

LT represents the final point of this continuum. Embedding YKL-40 within multidimensional predictive frameworks that combine biological, radiological, and clinical data would improve candidate selection, guide decisions on bridging therapy, and tailor post-transplant surveillance for recurrence and graft fibrosis. Harmonized multicenter registries are essential to establish dynamic thresholds, standardize assays, and validate performance across diverse patient populations.

Advances in computational science provide the tools needed to operationalize these concepts. Machine learning and artificial intelligence frameworks are well suited to model the nonlinear interactions between stromal activation, immune modulation, and tumor biology. By integrating real-time biomarker data with imaging, clinical variables, and tumor characteristics, these systems can generate continuously updated, individualized risk estimates. Such adaptive models move beyond static cut-offs to deliver risk-adapted care pathways, aligning surveillance intensity, bridging therapy, and adjuvant strategies with evolving tumor-host interactions. Through this integration, YKL-40 could progress from a descriptive read-out to a functional driver of precision hepatology and transplantation.

Therapeutic implications are also emerging as YKL-40 moves beyond its role as a biomarker. Experimental studies have shown that neutralizing YKL-40 with monoclonal antibodies can reduce angiogenesis, fibrosis, and tumor progression by interrupting IL-13Rα2-mediated signaling pathways [115,116]. In particular, the IL-13Rα2/TMEM219 complex has been identified as a critical mediator of YKL-40-driven MAPK and PI3K/AKT activation, linking this pathway to fibroblast and stellate cell activation [117]. Inhibition of this receptor complex could therefore prevent downstream profibrotic signaling, reducing HSC proliferation and collagen deposition. These strategies are conceptually similar to antifibrotic therapies targeting TGF-β or LOXL2 but uniquely focus on disrupting the feed-forward loop between YKL-40 and ECM remodeling. While no clinical trials have yet evaluated these approaches in viral hepatitis or LT, preclinical data provide a strong rationale for testing YKL-40 blockade as a novel therapeutic avenue to slow fibrosis progression, reduce HCC recurrence risk, and improve graft outcomes after transplantation.

## 5. Conclusions

The value of YKL-40 lies less in isolated performance metrics and more in its ability to signal a transition from morphology-driven assessment toward biologically adaptive models of care in liver transplantation. While current evidence shows consistent associations with fibrosis stage, tumor behavior, and graft outcomes, its adoption in clinical pathways remains limited. Progress will depend on robust validation in multicenter cohorts, harmonization of assay platforms, and clear definition of dynamic thresholds that can be interpreted across different disease stages and treatment states. Prospective integration of YKL-40 into transplant registries and biomarker-guided surveillance trials will be essential to determine whether it adds value beyond established scores and imaging. The next step is not to test YKL-40 in isolation but to embed it into algorithmic frameworks that combine stromal, tumor, and host signals. Such multidimensional approaches can support risk-adaptive strategies and bring precision medicine in transplantation closer to practice.

## Figures and Tables

**Figure 1 ijms-26-09584-f001:**
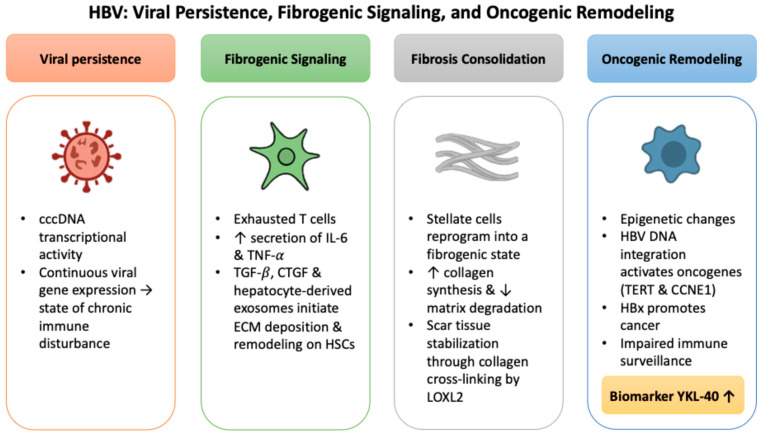
HBV-associated mechanisms of viral persistence, fibrogenic signaling, and oncogenic remodeling converging on ECM remodeling and YKL-40 induction.

**Figure 2 ijms-26-09584-f002:**
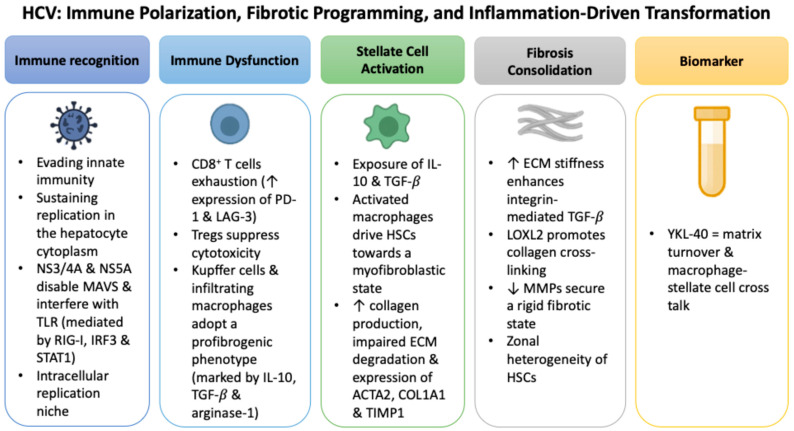
Sequential mechanisms of HCV-driven immune evasion, fibrotic programming, and inflammation-driven transformation leading to YKL-40 induction.

**Figure 3 ijms-26-09584-f003:**
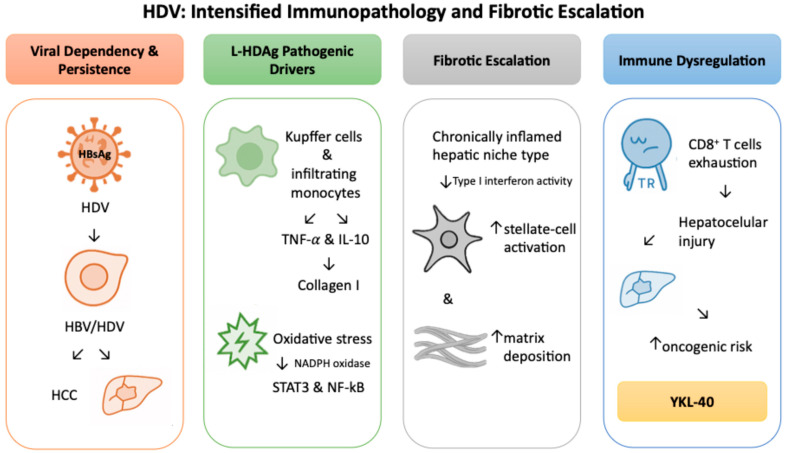
Intensified immunopathology and accelerated fibrotic progression in HBV/HDV co-infection leading to YKL-40 induction.

**Figure 4 ijms-26-09584-f004:**
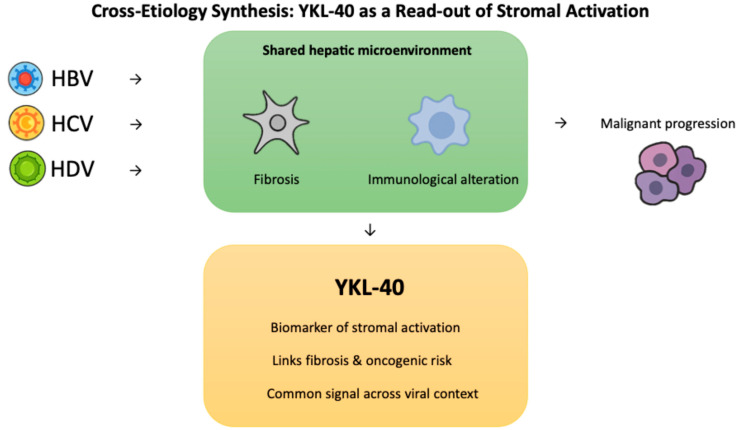
Cross-etiology convergence of HBV, HCV, and HDV on YKL-40 as a biomarker linking fibrosis and oncogenic risk.

**Figure 5 ijms-26-09584-f005:**
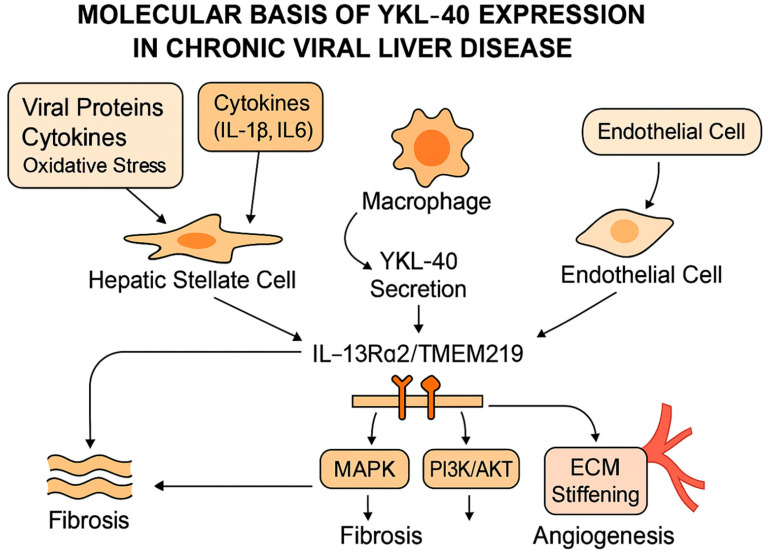
Molecular basis of YKL-40 expression linking viral triggers, cellular sources, and downstream profibrotic signaling.

**Figure 6 ijms-26-09584-f006:**
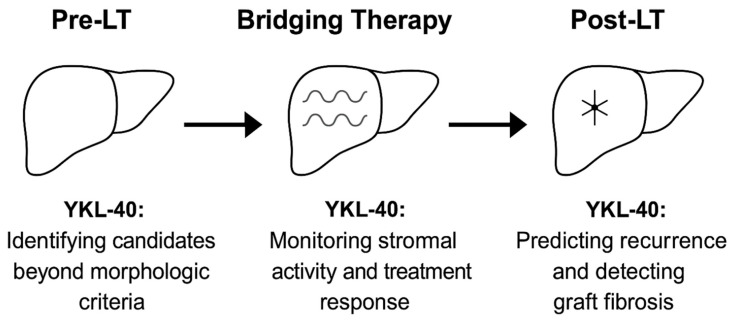
Overview of YKL-40 applications in LT, from pre-transplant eligibility to post-transplant surveillance.

**Table 1 ijms-26-09584-t001:** Biomarker classes relevant to viral hepatitis-associated liver disease and along the LT pathway: principal biology, clinical readouts, evidence in HBV/HCV/HDV, candidate LT decision points, and key caveats.

Biomarker Class	Biology/Principal Source	What the Marker Captures	Evidence in Viral Hepatitis and HCC (HBV/HCV/HDV)	Where It Can Inform the LT Pathway *	Key Caveats
AFP	Tumor-derived glycoprotein (oncofetal)	Tumor burden/secretory phenotype	Widely used in HBV/HCV HCC; imperfect sensitivity in early/non-secretory tumors	Pre-LT: selection models (e.g., AFP-based); Bridging: response tracking; Post-LT: recurrence surveillance adjunct	False positives (active hepatitis, pregnancy); non-secretors
DCP (PIVKA-II)	Abnormal prothrombin (des-γ-carboxylated)	Tumor biology; angiogenesis/ vitamin-K axis	Complements AFP in HBV/HCV; prognostic value	Pre-LT: MoRAL-type scores; Bridging: biology signal; Post-LT: recurrence risk adjunct	Affected by vitamin-K status/ warfarin; assay variability
GPC3	Oncofetal proteoglycan (tumor membrane/serum)	Tumor presence/ aggressiveness	Overexpressed in viral-related HCC; IHC and serum utility	Pre-LT: biology beyond size/number; Bridging: residual activity	Serum assays not standardized; not expressed in all HCC
OPN	Matricellular protein (hepatocytes, stroma, immune)	Inflammation, fibrosis, invasion	Elevated in HBV/HCV HCC; prognostic associations	Pre-LT: risk enrichment; Bridging: limited but plausible; Post-LT: prognosis (exploratory)	Non-specific; influenced by systemic inflammation
ctDNA	Tumor-derived cfDNA (mutational/ epigenetic)	Genomic/epigenetic alterations; MRD	HBV-HCC meta-analyses; increasing early-detection/ monitoring data	Pre-LT: biology beyond imaging (VET, MVI risk); Bridging: MRD after downstaging; Post-LT: molecular recurrence	Low tumor fraction in early disease; high-complexity assays
microRNA (cell-free/EV cargo)	Regulatory RNAs (hepatocytes, tumor, immune)	Pathway dysregulation; injury/oncogenic programs	HBV/HCV signatures (e.g., miR-122/21 panels)	Pre-LT: risk phenotyping; Post-LT: recurrence/ rejection (emerging)	Normalization, platform and pre-analytics; heterogeneity
EVs	Vesicles from liver/tumor/immune cells	Intercellular crosstalk; composite cargo (miRNA/protein)	Viral hepatitis cohorts; fibrosis/HCC signals	Pre-LT/Bridging: disease activity; Post-LT: rejection/ fibrosis (exploratory)	Isolation/ quantification not harmonized; specificity
YKL-40	HSCs, macrophages, endothelium, stressed hepatocytes	Stromal activation/fibrogenesis/angiogenic tone	Elevated with advanced fibrosis; prognostic in HCC; biologically coherent in HBV/HCV	Pre-LT: selection beyond morphology; Bridging: trajectory vs. biology; Post-LT: recurrence and graft fibrosis signals	Non-specific (inflammation, infection); age/genetic effects; assay variability

AFP: Alpha-fetoprotein; cfDNA: Cell-free DNA; ctDNA: Circulating tumor DNA; DCP: Des-gamma-carboxy prothrombin; EVs: Extracellular vesicles; EV cargo: Extracellular vesicle cargo; GPC3: Glypican-3; HBV: Hepatitis B virus; HCV: Hepatitis C virus; HDV: Hepatitis D virus; HCC: Hepatocellular carcinoma; IHC: Immunohistochemistry; LT: Liver transplantation; miRNA: MicroRNA; MoRAL: Model to predict tumor Recurrence After Living-donor liver transplantation; MRD: Minimal (molecular) residual disease; MVI: Microvascular invasion; OPN: Osteopontin; PIVKA-II: Protein induced by vitamin K absence or antagonist-II; Pre-LT: Pre-liver transplantation; Post-LT: Post-liver transplantation; VET: Viable enhancing tumor; YKL-40: Chitinase-3-like protein 1. * Pre-LT = candidacy and risk stratification; Bridging = response to downstaging/locoregional therapy; Post-LT = recurrence surveillance and graft injury/fibrosis.

**Table 2 ijms-26-09584-t002:** Comparative mechanisms of YKL-40 induction and function in HBV and HCV infection: from viral triggers to biomarker translation.

Pathogenic Axis	HBV-Associated Mechanisms	HCV-Associated Mechanisms	Shared Features
Viral trigger	HBx activates TGF-β/SMAD and IL-6/STAT3 signaling	Core and NS5A proteins induce ROS and NF-κB activation	Upregulation of IL-6, TGF-β, and VEGF
Fibrogenesis	cccDNA persistence sustains HSC activation, even in inactive carriers	ROS-driven HSC activation and progressive fibrotic remodeling	Collagen I and fibronectin accumulation with impaired ECM turnover
Immune modulation	Expansion of M2 macrophages and IL-10, dominant immune tolerance	T-cell exhaustion with PD-1^+^ CD8^+^ subsets	Recruitment of Tregs and MDSCs; promotion of angiogenesis
YKL-40 expression	Elevated even at low viremia, correlates with necroinflammatory activity	Persists in a subset after SVR, associated with residual fibrosis	Reflects active stromal remodeling and immunosuppressive microenvironment
Biomarker potential	Sensitive marker of fibrogenic activity in HCC-naïve patients	Identifies a high-risk phenotype post-SVR	Applicable to early HCC risk stratification and supportive in pre-transplant decision-making pathways

cccDNA: Covalently closed circular DNA; CD8+: Cluster of differentiation 8-positive; ECM: Extracellular matrix; HCC: Hepatocellular carcinoma; HBV: Hepatitis B virus; HBx: Hepatitis B X protein; HCV: Hepatitis C virus; HSC: Hepatic stellate cell; IL-6: Interleukin-6; IL-10: Interleukin-10; LT: Liver transplantation; M2: M2 macrophage phenotype; MDSCs: Myeloid-derived suppressor cells; NF-κB: Nuclear factor kappa-light-chain-enhancer of activated B cells; NS5A: Non-structural protein 5A; PD-1: Programmed cell death protein 1; ROS: Reactive oxygen species; SMAD: Small mothers against decapentaplegic (SMAD family of proteins); STAT3: Signal transducer and activator of transcription 3; SVR: Sustained virologic response; TGF-β: Transforming growth factor-beta; Tregs: Regulatory T cells; VEGF: Vascular endothelial growth factor; YKL-40: Chitinase-3-like protein 1.

**Table 3 ijms-26-09584-t003:** Diagnostic performance of YKL-40 in viral liver diseases: receiver operating characteristic (ROC)-based thresholds, sensitivity, and specificity in HBV and HCV cohorts.

Viral Etiology	First Author (Year)	Cohort/Setting	Diagnostic Endpoint *	AUC	Cut-Off (ng/mL)	Sensitivity (%)	Specificity (%)
**HBV**	Jiang (2020) [32]	CHB; biopsy-verified fibrosis	Significant fibrosis (F0-F1 vs. F2-F3)	0.970	68.75	95.2	89.7
**HBV**	Wang (2018) [33]	CHB; baseline biopsy cohort	Ishak ≥ F2	0.86	60.9	82	83
			Ishak ≥ F3	— (NR)	73.8	53	70
			Ishak ≥ F4	— (NR)	91.9	69	67
			Ishak ≥ F5	— (NR)	106.9	61	70
**HCV**	Saitou (2005) [55]	CHC; biopsy cohort	Significant fibrosis (F0-F1 vs. F2-F4)	0.809	186.4	78	81
			Cirrhosis (F4 vs. F0-F3)	0.795	284.8	80	71

CHB: Chronic hepatitis B; CHC: Chronic hepatitis C; HBV: Hepatitis B virus; HCV: Hepatitis C virus; NR: Not reported. * Fibrosis stages according to METAVIR: F0 = no fibrosis; F1 = portal fibrosis without septa; F2 = few septa; F3 = numerous septa without cirrhosis; F4 = cirrhosis. Ishak ≥F2, ≥F3, ≥F4, and ≥F5 represent progressively higher fibrosis thresholds for staging liver disease severity.

**Table 4 ijms-26-09584-t004:** YKL-40-associated tumor mechanisms and their clinical implications in pre-liver transplant HCC risk stratification.

Tumor-Promoting Mechanism	YKL-40-Associated Effect	Clinical Implication
Activation of invasive signaling	Activates PI3K/AKT and TGF-β pathways in cancer models	Indicates high-risk tumor phenotype beyond AFP/morphology
ECM remodeling	Upregulates MMP-9 expression, promoting ECM degradation	Reflects stromal remodeling undetectable by imaging
Loss of epithelial adhesion	Suppresses E-cadherin, facilitating epithelial–mesenchymal transition (EMT)	Correlates with dedifferentiation and early metastatic potential
Angiogenesis	Induces endothelial cell migration and tube formation independently of VEGF	May predict microvascular invasion and post-LT recurrence
Resistance to bridging therapy	Persistently elevated levels despite radiologic response suggest stromal or residual tumor activity	Supports re-evaluation of treatment response and LT candidacy

AFP: Alpha-fetoprotein; AKT: Protein kinase B; ECM: Extracellular matrix; EMT: Epithelial–mesenchymal transition; E-cadherin: Epithelial cadherin; LT: Liver transplantation; MMP-9: Matrix metalloproteinase-9; PI3K: Phosphoinositide 3-kinase; TGF-β: Transforming growth factor-beta; VEGF: Vascular endothelial growth factor; YKL-40: Chitinase-3-like protein 1.

## Data Availability

No new data were created or analyzed in this study. Data sharing is not applicable to this article.

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
