# Peer review of "YKL-40 in Virus-Associated Liver Disease: A Translational Biomarker Linking Fibrosis, Hepatocarcinogenesis, and Liver Transplantation"

_ijms, 2025, doi:10.3390/ijms26199584_

Round 1

Reviewer 1 Report

Comments and Suggestions for Authors

This manuscript summarizes the research progress on YKL-40 as a biomarker for viral-associated liver disease. The authors first describe the biological mechanisms and clinical relevance of YKL-40 in viral-associated liver fibrosis and liver cancer. They then describe the research progress of YKL-40 as a biomarker and summarize its value in liver transplant risk assessment. Overall, this review is relatively systematic, and I recommend it be revised. However, the introduction, research perspectives, and figures need improvement. I suggest that it be published after appropriate revisions. Specific comments are as follows:
1. The current introduction is too concise. Given the extensive research on liver disease-related biomarkers, not just conventional molecules like AFP and DCP, many novel molecules have also been reported. I suggest that the authors enrich the introduction.
2. In the introduction, I suggest that the authors highlight the research significance of viral-associated liver disease biomarkers.
3. This article lacks figures, only two tables. I suggest that the authors add one or two figures. This will enhance the text and facilitate a quick understanding of the YKL-40 regulatory network, which will also increase the readership of this study. 4. This article only provides an overview of future research directions for YKL-40 in liver transplantation, but does not discuss or outline overall viral liver diseases such as liver fibrosis, cirrhosis, and liver cancer.
5. It is recommended to add a chart summarizing the research progress of YKL-40 as a biomarker in different viral liver diseases, including information such as ROC diagnostic AUC values, threshold values, diagnostic specificity, and sensitivity.

Author Response

Reviewer #1

The current introduction is too concise. Given the extensive research on liver disease-related biomarkers, not just conventional molecules like AFP and DCP, many novel molecules have also been reported. I suggest that the authors enrich the introduction.

(Response)

Thank you for the suggestion. We have enriched the “Introduction” to contextualize contemporary biomarkers beyond AFP/DCP, namely glypican-3, osteopontin, circulating tumor DNA, microRNAs (including EV cargo), and extracellular vesicles, and to clarify the biological layers each interrogates. We also added Table 1 summarizing key features and LT use-cases across the viral hepatitis-HCC-LT continuum. All references have been updated to reflect current literature.

Page 2, line 71: Consequently, current work spans complementary biomarker classes, such as tumor-derived proteins (e.g., glypican-3 and osteopontin), liquid biopsy analytes (circulating tumor DNA and microRNAs, including extracellular vesicle cargo), and extracellular vesicles themselves, which probe oncoprotein expression, genomic/epigenetic alterations, and vesicle-mediated stromal-immune crosstalk [9-13]. An overview of conventional and emerging biomarkers across the viral hepatitis-HCC-LT continuum is summarized in Table 1.

Page 3, line 74: “Table 1. Biomarker classes relevant to viral hepatitis-associated liver disease and along the LT pathway: principal biology, clinical readouts, evidence in HBV/HCV/HDV, candidate LT deci-sion points, and key caveats.”

Biomarker class

Biology/principal source

What the marker captures

Evidence in viral hepatitis & HCC (HBV/HCV/HDV)

Where it can inform the LT pathway*

Key caveats

AFP

Tumor-derived glycoprotein (oncofetal)

Tumor burden/secretory phenotype

Widely used in HBV/HCV HCC; imperfect sensitivity in early/non-secretory tumors

Pre-LT: selection models (e.g., AFP-based); Bridging:response tracking; Post-LT: recurrence surveillance adjunct

False positives (active hepatitis, pregnancy); non-secretors

DCP (PIVKA-II)

Abnormal prothrombin (des-γ-carboxylated)

Tumor biology; angiogenesis/

vitamin-K axis

Complements AFP in HBV/HCV; prognostic value

Pre-LT: MoRAL-type scores; Bridging: biology signal; Post-LT:recurrence risk adjunct

Affected by vitamin-K status/

warfarin; assay variability

GPC3

Oncofetal proteoglycan (tumor membrane/serum)

Tumor presence/

aggressiveness

Overexpressed in viral-related HCC; IHC and serum utility

Pre-LT: biology beyond size/number; Bridging: residual activity

Serum assays not standardized; not expressed in all HCC

OPN

Matricellular protein (hepatocytes, stroma, immune)

Inflammation, fibrosis, invasion

Elevated in HBV/HCV HCC; prognostic associations

Pre-LT: risk enrichment; Bridging: limited but plausible; Post-LT:prognosis (exploratory)

Non-specific; influenced by systemic

inflammation

ctDNA

Tumor-derived cfDNA (mutational/

epigenetic)

Genomic/epigenetic alterations; MRD

HBV-HCC meta-analyses; increasing early-detection/

monitoring data

Pre-LT: biology beyond imaging (VET, MVI risk); Bridging: MRD after downstaging; Post-LT: molecular recurrence

Low tumor fraction in early disease; high-complexity assays

microRNA (cell-free/EV cargo)

Regulatory RNAs (hepatocytes, tumor, immune)

Pathway dysregulation; injury/oncogenic programs

HBV/HCV signatures (e.g., miR-122/21 panels)

Pre-LT: risk phenotyping; Post-LT: recurrence/

rejection (emerging)

Normalization, platform and pre-analytics; heterogeneity

EVs

Vesicles from liver/tumor/immune cells

Intercellular crosstalk; composite cargo (miRNA/protein)

Viral hepatitis cohorts; fibrosis/HCC signals

Pre-LT/Bridging:disease activity; Post-LT: rejection/

fibrosis (exploratory)

Isolation/

quantification not harmonized; specificity

YKL-40

HSCs, macrophages, endothelium, stressed hepatocytes

Stromal activation/fibrogenesis/angiogenic tone

Elevated with advanced fibrosis; prognostic in HCC; biologically coherent in HBV/HCV

Pre-LT: selection beyond morphology; Bridging: trajectory vs. biology; Post-LT:recurrence and graft fibrosis signals

Non-specific (inflammation, infection); age/genetic effects; assay variability

In the introduction, I suggest that the authors highlight the research significance of viral-associated liver disease biomarkers.

(Response)

Thank you for this constructive comment. We have revised the “Introduction“ to explicitly emphasize the research significance of viral-associated liver disease biomarkers by linking them to the global burden of viral hepatitis, their role in early detection and disease staging, and their potential impact on transplant decision-making. 

Page 2, line 65: “In virus-associated disease, where virologic control reduces inflammatory activity but does not abolish carcinogenic risk, biologically informative biomarkers are needed to refine risk-targeted surveillance, enable earlier detection and response assessment (including molecular residual disease), and complement morphologic criteria along the liver transplantation (LT) pathway (candidate selection, bridging/downstaging, post-LT surveillance).”

This article lacks figures, only two tables. I suggest that the authors add one or two figures. This will enhance the text and facilitate a quick understanding of the YKL-40 regulatory network, which will also increase the readership of this study.

(Response)

Thank you for this valuable suggestion. We have significantly enhanced the visual presentation of the manuscript. It now includes a total of six figures and four tables, in order to provide clear visual summaries of key concepts and mechanisms, and to facilitate a clearer understanding of the role of YKL-40 along the liver disease and transplantation continuum.

This article only provides an overview of future research directions for YKL-40 in liver transplantation, but does not discuss or outline overall viral liver diseases such as liver fibrosis, cirrhosis, and liver cancer.

(Response)

Thank you for this insightful comment. We have substantially revised Section 4.4. “Future Directions” to broaden its scope beyond LT. The revised section now provides a clear, progressive narrative addressing the full continuum of virus-associated liver disease, including fibrosis, cirrhosis, and HCC, before culminating in LT and computational applications.

Page 16, line 496: Future investigations will need to focus on translating YKL-40 from a descriptive biomarker into a clinically actionable tool across the spectrum of virus-associated liver disease. At earlier stages, longitudinal monitoring of YKL-40 offers a strategy to capture ongoing stromal remodeling and to assess fibrosis regression during antiviral or antifibrotic therapy. Large, prospective cohorts with standardized protocols are required to define biologically meaningful fluctuations, link them to histological and clinical outcomes, and establish its role in risk stratification and therapeutic planning.

As liver architecture becomes more disrupted, YKL-40 has the potential to signal transition toward decompensation before overt clinical manifestations arise. When incorporated into multivariable frameworks alongside established clinical variables, this biomarker could refine the prediction of complications such as variceal bleeding, ascites, and hepatic encephalopathy. Aligning surveillance and preventive interventions with molecular disease activity rather than static scores would enable timelier and more efficient use of resources.

Persistent stromal activation also contributes to tumor initiation, progression, and recurrence, even when viral replication is well controlled. Future efforts should explore integrating YKL-40 into composite biomarker signatures that merge stromal, immune, and tumor-derived signals with advanced imaging and circulating markers such as ctDNA and microRNAs. Such models could enhance early detection, recurrence prediction, and post-treatment monitoring, particularly for AFP-negative phenotypes and patients under surveillance following curative-intent therapies.

LT represents the final point of this continuum. Embedding YKL-40 within multidimensional predictive frameworks that combine biological, radiological, and clinical data would improve candidate selection, guide decisions on bridging therapy, and tailor post-transplant surveillance for recurrence and graft fibrosis. Harmonized multicenter registries are essential to establish dynamic thresholds, standardize assays, and validate performance across diverse patient populations.

Advances in computational science provide the tools needed to operationalize these concepts. Machine learning and artificial intelligence frameworks are well suited to model the nonlinear interactions between stromal activation, immune modulation, and tumor biology. By integrating real-time biomarker data with imaging, clinical variables, and tumor characteristics, these systems can generate continuously updated, individualized risk estimates. Such adaptive models move beyond static cut-offs to deliver risk-adapted care pathways, aligning surveillance intensity, bridging therapy, and adjuvant strategies with evolving tumor-host interactions. Through this integration, YKL-40 could progress from a descriptive read-out to a functional driver of precision hepatology and transplantation.”

It is recommended to add a chart summarizing the research progress of YKL-40 as a biomarker in different viral liver diseases, including information such as ROC diagnostic AUC values, threshold values, diagnostic specificity, and sensitivity.

(Response)

Thank you for this excellent suggestion. We have carefully reviewed the literature and identified studies that reported ROC AUC values, cut-off thresholds, sensitivity, and specificity for YKL-40 in chronic viral hepatitis cohorts. Based on these data, we created a new table (Table 3) summarizing the diagnostic performance of YKL-40 in both HBV and HCV populations.

Page 13, line 409: “To provide a comprehensive overview of its diagnostic performance, Table 3 summarizes key studies reporting receiver operating characteristic (ROC) values, optimal cut-off thresholds, and corresponding sensitivity and specificity of YKL-40 in HBV and HCV cohorts.

Table 3. Diagnostic performance of YKL-40 in viral liver diseases: receiver operating character-istic (ROC)-based thresholds, sensitivity, and specificity in HBV and HCV cohorts.”

Viral Etiology

First Author (Year)

Cohort/Setting

Diagnostic endpoint*

AUC

Cut-off (ng/mL)

Sensitivity (%)

Specificity (%)

HBV

Jiang (2020) [32]

CHB; biopsy-verified fibrosis

Significant fibrosis (F0-F1 vs F2-F3)

0.970

68.75

95.2

89.7

HBV

Wang (2018) [33]

CHB; baseline biopsy cohort

Ishak ≥F2

0.86

60.9

82

83

Ishak ≥F3

— (NR)

73.8

53

70

Ishak ≥F4

— (NR)

91.9

69

67

Ishak ≥F5

— (NR)

106.9

61

70

HCV

Saitou (2005)

CHC; biopsy cohort

Significant fibrosis (F0-F1 vs F2-F4)

0.809

186.4

78

81

Cirrhosis (F4 vs F0-F3)

0.795

284.8

80

71

Reviewer 2 Report

Comments and Suggestions for Authors

Journal: IJMS (ISSN 1422-0067)

Manuscript ID: ijms-3868333

Type: Review

Title: YKL-40 in Virus-Associated Liver Disease: A Translational Biomarker Linking Fibrosis, Hepatocarcinogenesis, and Liver Transplantation

This manuscript is clear, well-organized, and timely. It does an excellent job of integrating the virology, molecular mechanisms, and clinical aspects of YKL-40, emphasizing its role in liver fibrosis, liver cancer, and transplant outcomes. One of its main advantages is that it combines basic science with practical clinical insights, making it useful for both researchers and doctors. With a few suggested figures and minor editorial changes, this review could make an important contribution to the field of liver disease biomarkers.

1: A graphical abstract that highlights the range of YKL-40 involvement from fibrosis to oncogenesis and transplantation outcomes would be a good addition for the authors to make in order to help readers better understand the idea.

2: The authors ought to think about including a schematic diagram that shows how the mechanisms of the viruses (HCV–NS proteins, HDV–L-HDAg, and HBV–HBx) converge on YKL-40 induction.

3: The mechanism focuses on immune evasion, but it doesn't explain how HCV proteins specifically cause YKL-40.

 4: Lines 117-133: Discuss a little more about direct viral–host interactions, such as YKL-40 promoter regulation and NS3/NS5A–STAT3 activation.

5: Lines 187-193: The authors admit that there is currently no direct evaluation of YKL-40 in HDV. Provide examples of any experimental approaches (clinical cohorts, transcriptomics in HDV-infected hepatocytes).

6: Lines 15-160: Fibrosis is associated with circulating YKL-40, but feedback loops between HSCs, ECM stiffening, and YKL-40 are not explained.YKL-40 → IL-13Rα2 → MAPK/PI3K → collagen deposition by adding a signaling model.

7: Lines 428–437: Therapeutic targeting is not discussed (particularly throughout).  Given how important YKL-40 is, might results change if it is blocked by antibodies or IL-13Rα2 antagonists? Include a paragraph discussing the therapeutic implications.

Author Response

Reviewer #2

A graphical abstract that highlights the range of YKL-40 involvement from fibrosis to oncogenesis and transplantation outcomes would be a good addition for the authors to make in order to help readers better understand the idea.

(Response)

Thank you for this valuable suggestion. We have created and added a new graphical abstract to visually depict the continuum of virus-associated liver disease, highlighting how YKL-40 links chronic viral hepatitis-driven fibrosis with hepatocarcinogenesis and key liver transplantation decision points (eligibility, bridging therapy, post-LT surveillance).

Page 2, line 46: Graphical abstract. YKL-40 as a translational biomarker bridging chronic viral hepatitis, fibrosis progression, hepatocarcinogenesis, and liver transplantation outcomes.

The authors ought to think about including a schematic diagram that shows how the mechanisms of the viruses (HCV–NS proteins, HDV–L-HDAg, and HBV–HBx) converge on YKL-40 induction.

(Response)

Thank you for highlighting this point. We have created a new figure (Figure 4) that schematically depicts how HBV, HCV, and HDV activate distinct profibrotic pathways that ultimately converge on YKL-40 induction.

Page 8, line 253: “The convergent and virus-specific pathways underlying YKL-40 induction are de-picted in Figure 4 and summarized in Table 2, highlighting how HBV, HCV, and HDV engage distinct viral drivers that ultimately activate overlapping profibrotic and in-flammatory cascades.

Figure 4. Cross-etiology convergence of HBV, HCV, and HDV on YKL-40 as a biomarker linking fibrosis and oncogenic risk.”

The mechanism focuses on immune evasion, but it doesn't explain how HCV proteins specifically cause YKL-40.

(Response)

Thank you for emphasizing this important aspect. Section 2.2 “HCV: Immune Polarization, Fibrotic Programming, and Inflammation-Driven Transformation” has been revised to clarify the mechanistic link between HCV viral proteins and YKL-40 expression. NS5A and core proteins are now described as inducing oxidative stress through NADPH oxidases and mitochondrial pathways while simultaneously activating NF-κB and STAT3 signaling.

Page 6, line 160: “Beyond immune evasion, HCV viral proteins actively drive profibrogenic signaling. Experimental studies have shown that NS5A and core proteins induce oxidative stress through NADPH oxidases and mitochondrial pathways, while simultaneously activating NF-κB and STAT3 signaling [49]. These pathways serve as established transcriptional activators of YKL-40, providing a mechanistic link between HCV protein activity and YKL-40 upregulation [49].”

Lines 117-133: Discuss a little more about direct viral–host interactions, such as YKL-40 promoter regulation and NS3/NS5A–STAT3 activation.

(Response)

Thank you for this insightful comment. Section 2.2 “HCV: Immune Polarization, Fibrotic Programming, and Inflammation-Driven Transformation” has been expanded to include a description of direct viral-host interactions. The revised text highlights that the YKL-40 promoter contains functional binding sites for NF-κB and STAT3, and that HCV NS3 and NS5A proteins sustain STAT3 activation through prolonged phosphorylation.

Page 6, line 166: The YKL-40 promoter contains functional response elements for both NF-κB and STAT3, rendering it highly responsive to these signaling cascades [50]. HCV NS3 and NS5A proteins sustain STAT3 activation through prolonged phosphorylation, directly enhancing YKL-40 transcription and linking viral replication to persistent stromal acti-vation and fibrogenesis [50].”

Lines 187-193: The authors admit that there is currently no direct evaluation of YKL-40 in HDV. Provide examples of any experimental approaches (clinical cohorts, transcriptomics in HDV-infected hepatocytes).

(Response)

Thank you for this constructive suggestion. Section 2.3. “HDV: Intensified Immunopathology and Fibrotic Escalation“ has been expanded to propose experimental strategies that could address this knowledge gap. These include multi-omics approaches, such as single-cell transcriptomics of HDV-infected hepatocytes or stellate cells, to map YKL-40 expression and regulatory networks, and clinical cohort studies with paired serum and liver biopsy data to assess its diagnostic and prognostic potential.

Page 8, line 235: “Future research could employ multi-omics approaches, such as single-cell transcriptomics of HDV-infected hepatocytes or stellate cells, to identify cellular sources and regulatory networks driving YKL-40 expression. In parallel, clinical cohort studies with paired serum and liver biopsy data could define the diagnostic and prognostic performance of YKL-40 in HBV/HDV coinfection, establishing its role as a potential biomarker across disease stages.”

Lines 15-160: Fibrosis is associated with circulating YKL-40, but feedback loops between HSCs, ECM stiffening, and YKL-40 are not explained.YKL-40 → IL-13Rα2 → MAPK/PI3K → collagen deposition by adding a signaling model.

(Response)

Thank you for pointing out this important mechanistic link. Section 3.1 has been expanded to describe how YKL-40 actively contributes to a profibrotic feedback cycle. The revised text now explains that binding of YKL-40 to IL-13Rα2/TMEM219 activates MAPK and PI3K/AKT pathways, promoting HSC proliferation and collagen deposition. Progressive ECM stiffening further enhances stellate cell activation and YKL-40 secretion, creating a self-sustaining loop that accelerates fibrosis.

Page 10, line 293: “In addition to being a marker of fibrosis, YKL-40 actively participates in a positive feedback loop that amplifies fibrogenesis. Upon secretion, YKL-40 binds to IL-13Rα2 in complex with TMEM219, triggering downstream MAPK and PI3K/AKT signaling cascades. These pathways promote HSC proliferation and collagen synthesis, reinforcing ECM deposition and stiffening. Increasing matrix stiffness in turn sustains stellate cell activation and further upregulates YKL-40 expression, creating a self-perpetuating cycle of fibrosis progression [82,83]. This mechanistic loop explains why circulating YKL-40 levels closely track disease severity and remain elevated even when upstream injury markers, such as ALT, return to normal.”

Lines 428–437: Therapeutic targeting is not discussed (particularly throughout).  Given how important YKL-40 is, might results change if it is blocked by antibodies or IL-13Rα2 antagonists? Include a paragraph discussing the therapeutic implications.

(Response)

Thank you for this thoughtful suggestion. We have expanded Section 4.4, Future Directions, to include a dedicated paragraph discussing therapeutic implications. This addition outlines experimental evidence demonstrating that YKL-40 neutralization with monoclonal antibodies can reduce angiogenesis, fibrosis, and tumor progression. It also highlights the role of the IL-13Rα2/TMEM219 receptor complex as a key mediator of downstream MAPK and PI3K/AKT activation, linking YKL-40 signaling to fibroblast and stellate cell activity.

Page 17, line 533: “Therapeutic implications are also emerging as YKL-40 moves beyond its role as a biomarker. Experimental studies have shown that neutralizing YKL-40 with monoclonal antibodies can reduce angiogenesis, fibrosis, and tumor progression by interrupting IL-13Rα2-mediated signaling pathways [115,116]. In particular, the IL-13Rα2/TMEM219 complex has been identified as a critical mediator of YKL-40-driven MAPK and PI3K/AKT activation, linking this pathway to fibroblast and stellate cell activation [117]. Inhibition of this receptor complex could therefore prevent downstream profibrotic signaling, reducing HSC proliferation and collagen deposition. These strategies are conceptually similar to antifibrotic therapies targeting TGF-β or LOXL2 but uniquely focus on disrupting the feed-forward loop between YKL-40 and ECM remodeling. While no clinical trials have yet evaluated these approaches in viral hepatitis or LT, preclinical data provide a strong rationale for testing YKL-40 blockade as a novel therapeutic avenue to slow fibrosis progression, reduce HCC recurrence risk, and improve graft outcomes after transplantation.”

Reviewer 3 Report

Comments and Suggestions for Authors

Recommendation: Reconsider after major revisions

Comments:

In this manuscript, the review article,on which the title “YKL-40 in Virus-Associated Liver Disease: A Translational Bi- 2 omarker Linking Fibrosis, Hepatocarcinogenesis, and Liver 3 Transplantation  could be reconsider,but after major revisions as followings:

  1. Firstly, the conclusion on whichYKL-40 in Virus-Associated Liver Disease: A Translational Bi- 2 omarker Linking Fibrosis, Hepatocarcinogenesis, and Liver 3 Transplantationshould be described and summarized with graphic abstract in clearly and simplified words. Suggestions to be added the clearly graphic abstract again, if possible.
  2. And also in the 2.1section listed in the text, the paragraph HBV: Viral Persistence, Fibrogenic Signaling, and Oncogenic Remodeling”should also be described and summarized with the clearly figures and diagrams in much more simplified words. Suggestions to be added some clearly figures and diagrams or tables, if possible!
  3. The same suggestions in 2.2 section, 2.3 and 2.4 sections, if provided some clearly figures and diagrams, that will be fine.
  4. In the 3.1 section, suggestions to be provided the clearly Figs and Diagrams or structure expressions for themolecular basis of YKL-40 Expression with clearly in much more simplified words 
  5. Lastly, suggestions in section: “ Pre- and Post-Transplant Applications of YKL-40 in HCC: From Eligi- 341 bility to Risk Stratiffcationif provided the clearly figures and diagrams in simplified words, that also will be fine.

Author Response

Reviewer #3

Firstly, the conclusion on which“YKL-40 in Virus-Associated Liver Disease: A Translational Bi- 2 omarker Linking Fibrosis, Hepatocarcinogenesis, and Liver 3 Transplantation”should be described and summarized with graphic abstract in clearly and simplified words. Suggestions to be added the clearly graphic abstract again, if possible.

(Response)

Thank you for this valuable suggestion. We have created and added a new graphical abstract to visually depict the continuum of virus-associated liver disease, highlighting how YKL-40 links chronic viral hepatitis-driven fibrosis with hepatocarcinogenesis and key liver transplantation decision points (eligibility, bridging therapy, post-LT surveillance).

Page 2, line 46: Graphical abstract. YKL-40 as a translational biomarker bridging chronic viral hepatitis, fibrosis progression, hepatocarcinogenesis, and liver transplantation outcomes.

And also in the 2.1 section listed in the text, the paragraph “HBV: Viral Persistence, Fibrogenic Signaling, and Oncogenic Remodeling”should also be described and summarized with the clearly figures and diagrams in much more simplified words. Suggestions to be added some clearly figures and diagrams or tables, if possible!

(Response)

Thank you for pointing out the importance of visually summarizing this section. We have added a new figure (Figure 1) to Section 2.1, “HBV: Viral Persistence, Fibrogenic Signaling, and Oncogenic Remodeling“, which provides a simplified overview of viral persistence, fibrogenic signaling, and oncogenic remodeling, illustrating how these pathways converge on ECM remodeling and YKL-40 induction.

Page 5, line 146: “The key processes of viral persistence, fibrogenic signaling, and oncogenic remodeling in HBV are summarized in Figure 1.

Figure 1. HBV-associated mechanisms of viral persistence, fibrogenic signaling, and oncogenic remodeling converging on ECM remodeling and YKL-40 induction.”

The same suggestions in 2.2 section, 2.3 and 2.4 sections, if provided some clearly figures and diagrams, that will be fine.

(Response)

Thank you for pointing out the importance of visually summarizing this section. We have added a new figure (Figure 2) to Section 2.2, “HCV: Immune Polarization, Fibrotic Programming, and Inflammation-Driven Transformation”, which provides a simplified overview of sequential immune dysfunction, fibrotic remodeling, and inflammation-driven transformation leading to YKL-40 induction.

Page 6, line 200: “Figure 2 summarizes the sequence of immune dysfunction, fibrotic remodeling, and YKL-40 induction in HCV-associated liver disease.

Figure 2. Sequential mechanisms of HCV-driven immune evasion, fibrotic programming, and inflamma-tion-driven transformation leading to YKL-40 induction.”

(Response)

Thank you for highlighting the value of adding visual summaries. We have created a new figure (Figure 3) for Section 2.3, “HDV: Intensified Immunopathology and Fibrotic Escalation”, to illustrate how HBV/HDV co-infection accelerates fibrotic progression and enhances inflammatory signaling, setting the stage for YKL-40 upregulation and increased oncogenic risk.

Page 8, line 240: “Figure 3 provides a visual summary of the intensified immunopathology and rapid fi-brotic escalation characteristic of HBV/HDV coinfection, illustrating how L-HDAg-driven immune dysregulation, oxidative stress, and stellate-cell activation converge on YKL-40 induction and heightened oncogenic risk.

Figure 3. Intensified immunopathology and accelerated fibrotic progression in HBV/HDV co-infection leading to YKL-40 induction.”

(Response)

Thank you for suggesting a visual representation for this section. We have added a new figure (Figure 4) to Section 2.4, “Cross-Etiology Synthesis: YKL-40 as a Read-out of Stromal Activation”, which integrates and contrasts the virus-specific and convergent pathways by which HBV, HCV, and HDV activate stromal remodeling and YKL-40 expression.

Page 8, line 253: “The convergent and virus-specific pathways underlying YKL-40 induction are de-picted in Figure 4 and summarized in Table 2, highlighting how HBV, HCV, and HDV engage distinct viral drivers that ultimately activate overlapping profibrotic and in-flammatory cascades.

Figure 4. Cross-etiology convergence of HBV, HCV, and HDV on YKL-40 as a biomarker linking fibrosis and oncogenic risk.”

In the 3.1 section, suggestions to be provided the clearly Figs and Diagrams or structure expressions for the molecular basis of YKL-40 Expression with clearly in much more simplified words

(Response)

Thank you for this helpful suggestion. We have added a new figure (Figure 5) to Section 3.1, “Molecular Basis of YKL-40 Expression”, to visually summarize the upstream triggers, cellular sources, and downstream profibrotic signalling pathways involved in YKL-40 expression. This schematic provides a simplified overview of how viral proteins, cytokines, and oxidative stress converge to drive YKL-40 secretion and sustain fibrosis through a positive feedback loop.

Page 10, line 300: “Figure 5 illustrates these interconnected mechanisms, showing how viral proteins, cytokines, and oxidative stress activate multiple liver cell populations to secrete YKL-40, which in turn drives fibrosis, angiogenesis, and ECM stiffening through MAPK and PI3K/AKT signaling pathways.

Figure 5. Molecular basis of YKL-40 expression linking viral triggers, cellular sources, and downstream profibrotic signaling.”

Lastly, suggestions in section: “ Pre- and Post-Transplant Applications of YKL-40 in HCC: From Eligi- 341 bility to Risk Stratiffcation”if provided the clearly figures and diagrams in simplified words, that also will be fine.

(Response)

Thank you for this thoughtful suggestion. We have created and added a new schematic figure (Figure 6) to Section 4, “Pre- and Post-Transplant Applications of YKL-40 in HCC: From Eligibility to Risk Stratification”, which visually summarizes how YKL-40 contributes to decision-making along the liver transplantation pathway.

Page 14, line 442: “Figure 6 provides a visual summary of how YKL-40 contributes to decision-making at different stages of the LT pathway, from pre-transplant eligibility to post-transplant surveillance.

Figure 6. Overview of YKL-40 applications in LT, from pre-transplant eligibility to post-transplant surveillance.”

Round 2

Reviewer 1 Report

Comments and Suggestions for Authors

The revised manuscript has addressed my concerns, and I recommend its acceptance for publication.

Reviewer 3 Report

Comments and Suggestions for Authors

Recommendation: Accept in present form and does not require further revisions.
Comments:
In this manuscript, the review article,on which the title “YKL-40 in Virus-Associated Liver
Disease: A Translational Bi- 2 omarker Linking Fibrosis, Hepatocarcinogenesis, and Liver 3
Transplantation ” could be acceptable, and no further revisions required.

Best wishes

Dr. Meiluo
HFUT
Hefei, anhui, china, 230009
